# Carbene organic catalytic planar enantioselective macrolactonization

Xiaokang Lv[1], Fen Su[1], Hongyan Long[1], Fengfei Lu[1], Yukun Zeng[1], Minghong Liao[1], Fengrui Che[1], Xingxing Wu [1] ✉ & Yonggui Robin Chi [1,2] ✉

Macrolactones exhibit distinct conformational and configurational properties and are widely found in natural products, medicines, and agrochemicals. Up to now, the major effort for macrolactonization is directed toward identifying suitable carboxylic acid/alcohol coupling reagents to address the challenges associated with macrocyclization, wherein the stereochemistry of products is usually controlled by the substrate's inherent chirality. It remains largely unexplored in using catalysts to govern both macrolactone formation and stereochemical control. Here, we disclose a non-enzymatic organocatalytic approach to construct macrolactones bearing chiral planes from achiral substrates. Our strategy utilizes N-heterocyclic carbene (NHC) as a potent acylation catalyst that simultaneously mediates the macrocyclization and controls planar chirality during the catalytic process. Macrolactones varying in ring sizes from sixteen to twenty members are obtained with good-to-excellent yields and enantiomeric ratios. Our study shall open new avenues in accessing macrolactones with various stereogenic elements and ring structures by using readily available small-molecule catalysts.

Macrolactones, cyclic carboxylic esters with over twelve-membered rings, are broadly present in natural and synthetic functional molecules[1]. Representative examples of bioactive macrolactones include medicinally important macrolide antibiotics such as erythromycin, and agrochemicals like avermectin and Spinosad (Fig. 1A)[2–4]. The unique conformational (and configurational) properties posed by the macrocyclic structures are critical for these molecules to display the right bioactivities[5,6]. Therefore, the preparation of macrolactones has received significant attention over the past decades[1,7], and new synthetic methods continue to emerge in recent years[8,9]. One class of such methods starts from substrates with carboxylic ester moieties pre-installed on the main chains and uses various transformations (such as those based on transition metal-catalyzed bond formations) to close the respective macrocycles[10–13]. The other type of methods, primarily a classic strategy, relies on the formation of carboxylic esters from the corresponding carboxylic acids and alcohols as the ring-closing step (Fig. 1B)[14–18]. Indeed, this

lactonization method still constitutes as the most prevalent, reliable approach, and holds clear promise especially since the selective acyl transfer reactions have been well explored through activation of acyl donors by both small-molecule[19–23] and enzyme catalysts[24–26]. In the past, studies on macrolactonization mainly focused on developing new carboxylic acid/alcohol coupling reagents and methods to ensure the lactone formations, in which the stereochemical course is mostly controlled by the inherent pre-existing chirality of substrates. There are fewer studies on using catalysts to simultaneously govern the lactonization reaction and stereochemical controls. In 2020, Collins and co-workers showed that the reaction between di-carboxylic acids (tethered with aliphatic linkers) and ortho-substituted benzylic diols can be mediated by the Candida antarctica lipase B (CALB) enzyme catalyst to form macrolactones (Fig. 1C)[27]. Due to rotational constraints posed by the macrocycles, the macrolactone products feature planar chirality and are obtained with excellent enantioselectivities under the control of the CALB enzyme catalyst. Additionally, it is noteworthy that

[1]National Key Laboratory of Green Pesticide, Key Laboratory of Green Pesticide and Agricultural Bioengineering, Ministry of Education, Guizhou University, Huaxi District, Guiyang 550025, China. [2]School of chemistry, chemical engineering, and biotechnology, Nanyang Technological University, Singapore 637371, Singapore. ✉e-mail: wuxx@gzu.edu.cn; robinchi@ntu.edu.sg

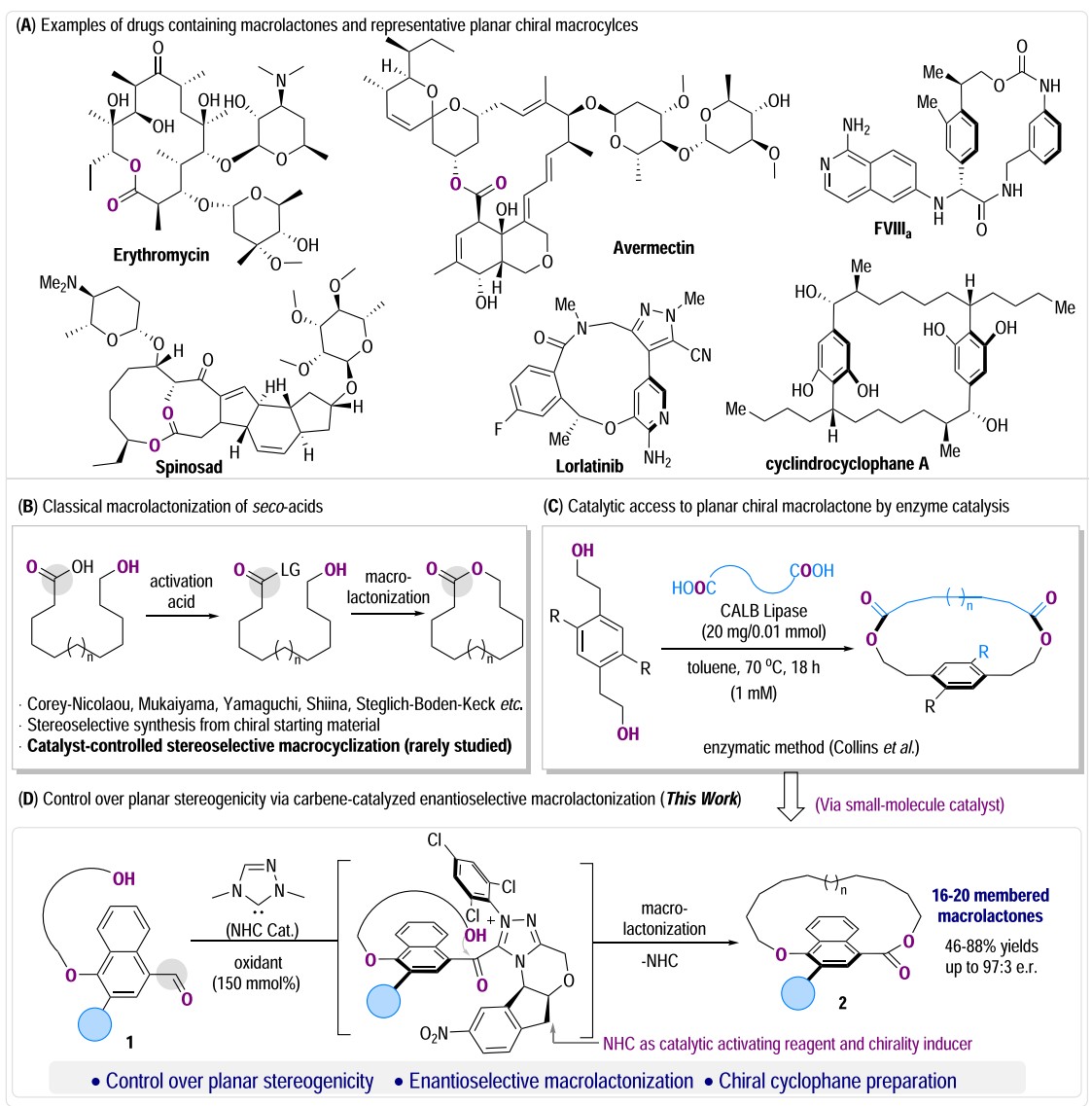

**Fig. 1 | Importance of macrolactones and the enantioselective synthesis.**
**A** Examples containing macrolactone motif and representative planar chiral mac-rocyles; **B** Typical synthetic strategy via macrolactonization; **C** Preparation of planar chiral macrolactones by enzyme catalysis; **D** Our proposed carbene orga-nocatalytic strategy for planar enantioselective macrolactonization.

chiral frameworks with planar stereogenicity are not only of high interest in their distinct molecular chirality, but also broadly present in natural products and utilized in asymmetric catalysis and materials (Fig. 1A)[28–30]. Despite this elegant work[27], examples to address the long-standing challenge of catalytic approach toward planar chiral cyclo-phane molecules remains largely unexplored[31–38].

We are interested in exploring N-heterocyclic carbene (NHC) as a small-molecule catalyst to address synthetic challenges in complex molecules[39]. NHC is, in principle, a class of excellent acylation catalysts that can offer multiple handles to modulate reactivity and chemo/regio/stereo-selectivities[40–45]. In this work, we disclose a non-enzymatic organocatalytic strategy for efficient access to planar chiral macro-lactones (Fig. 1D). The reaction of the NHC catalyst with aldehyde moiety of the bifunctional hydroxyl aldehyde substrate **1** under oxi-dative conditions effectively leads to an NHC-bound acyl azolium intermediate[46–49]. This intermediate then reacts with the alcohol moi-ety of the substrate to close the lactone ring. The pro-chiral arene planes from substrate **1** in the intermediate are well differentiated by stereo-control of the chiral NHC catalyst, leading to the planar chiral macrolactones **2** with good yields and excellent enantiomeric ratios. In the long round, our study shall open new avenues in accessing mac-rolactones with various chirality styles and ring structures by using readily available small-molecule catalysts.

## Results and discussion

We commenced our investigation with the expeditious preparation of acyclic substrate **1a** (Fig. 2). Substituted naphthol aldehyde **4a** was easily accessed through a reaction sequence involving a formylation and bromination process from commercially available 1-naphthol (**3a**). A side chain was subsequently installed with **5a** via a Mitsunobu reaction, leading to the desired model substrate **1a** through mild deprotection of the TBS group upon treatment with TBAF.

With the acyclic aldehyde **1a** in hand, we set out to study the carbene-catalyzed enantioselective macrocyclization (Fig. 3). Gratify-ingly, the reaction enabled by achiral NHC **A** with K$_2$CO$_3$ in toluene at 100 °C gave rise to the desired product **2a** in 35% yield, revealing the feasibility of NHC-catalyzed macrolactonization transformation. Var-ious chiral carbene catalysts **B-H** were then carefully screened to

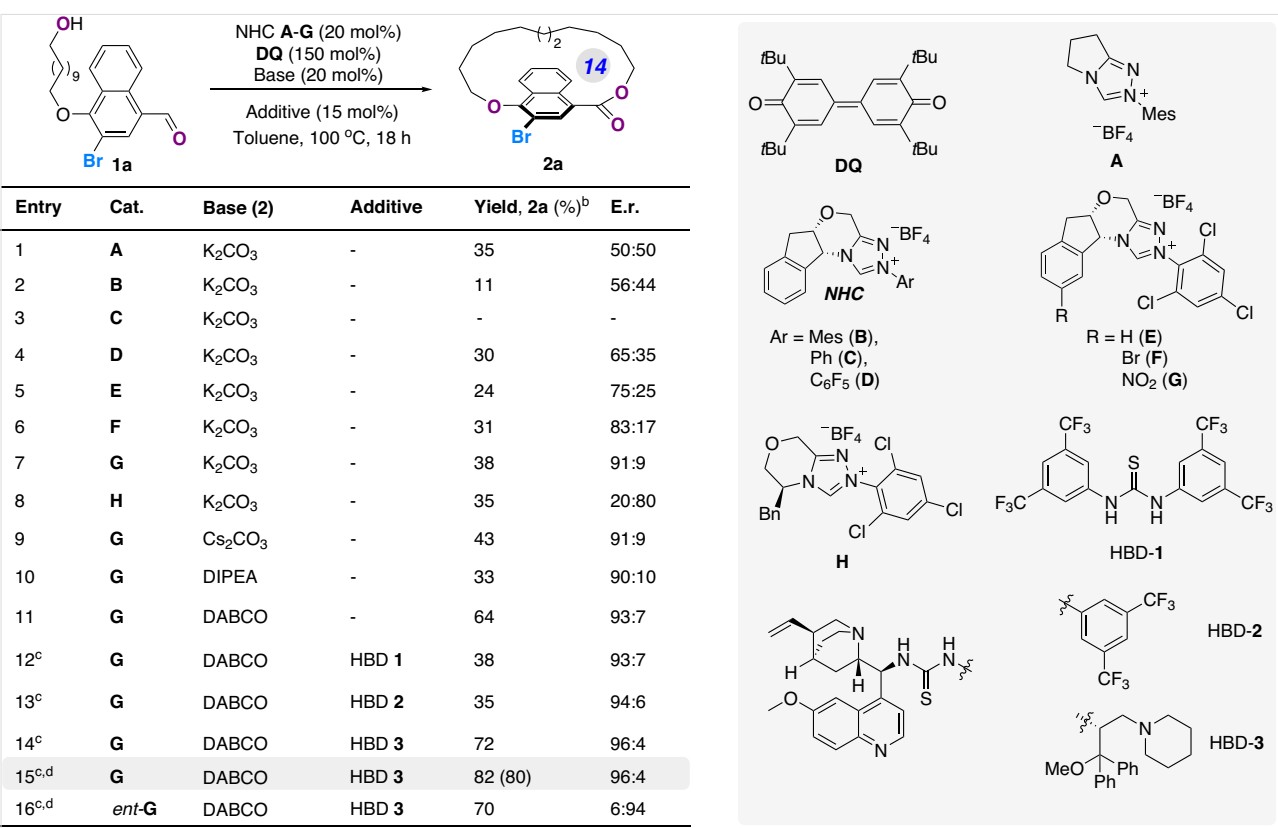

**Fig. 2 | Synthetic protocol of substrate 1a.** (1) Titanium tetrachloride, dichloro(methoxy)methane, THF, 0 °C to rt, 10 min; 82% yield; (2) Diisopropylamine, NBS, CH₂Cl₂, 45 °C, 16 h; 62% yield; (3) DIAD, PPh₃, **5a**, THF, 0 °C to rt, 12 h; 82% yield; (4) TBAF, THF, rt, 12 h; 60% yield. NBS N-bromosuccinimide, TBS *tert*-butyldimethylsilyl, TBAF tetrabutylammonium fluoride, DIAD diisopropyl azodicarboxylate.

| Entry | Cat. | Base (2) | Additive | Yield, 2a (%)[b] | E.r. |
|---|---|---|---|---|---|
| 1 | **A** | K₂CO₃ | - | 35 | 50:50 |
| 2 | **B** | K₂CO₃ | - | 11 | 56:44 |
| 3 | **C** | K₂CO₃ | - | - | - |
| 4 | **D** | K₂CO₃ | - | 30 | 65:35 |
| 5 | **E** | K₂CO₃ | - | 24 | 75:25 |
| 6 | **F** | K₂CO₃ | - | 31 | 83:17 |
| 7 | **G** | K₂CO₃ | - | 38 | 91:9 |
| 8 | **H** | K₂CO₃ | - | 35 | 20:80 |
| 9 | **G** | Cs₂CO₃ | - | 43 | 91:9 |
| 10 | **G** | DIPEA | - | 33 | 90:10 |
| 11 | **G** | DABCO | - | 64 | 93:7 |
| 12[c] | **G** | DABCO | HBD 1 | 38 | 93:7 |
| 13[c] | **G** | DABCO | HBD 2 | 35 | 94:6 |
| 14[c] | **G** | DABCO | HBD 3 | 72 | 96:4 |
| 15[c,d] | **G** | DABCO | HBD 3 | 82 (80) | 96:4 |
| 16[c,d] | *ent*-**G** | DABCO | HBD 3 | 70 | 6:94 |

**Fig. 3 | Optimization of the carbene-catalyzed enantioselective macrocyclization.** [a] The reactions were performed with **1a** (6.3 mg, 0.015 mmol, 1.0 equiv.), NHC **A-H** (20 mol%), DQ (9.2 mg, 150 mol%), and base (20 mol%) in toluene (1 mM) under N₂ atmosphere at 100 °C for 12 h; [b] Yields of **2a** were determined via ¹H NMR analysis with 1,3,5-trimethoxybenzene as an internal standard; Isolated yield in the parenthesis; [c] 15 mol% of additive was used; [d] A mixed solvent (toluene:*n*-heptane = 11:9, 1 mM) was used. DIEA *N, N*-diisopropylethylamine, DABCO triethylenediamine, DQ 3,3′,5,5′-tetra-tert-butyldiphenoquinone, HBD hydrogen bond donor.

explore their capability for stereo-control over planar stereogenicity of the macrocyclic product **2a**. Whereas indanol-based catalyst **D** afforded the product in modest enantioselectivity (entries 2–4), we were pleased to find that N-2,4,6-trichlorophenyl substituted NHC **E** was superior to give the product **2a** in a promising 75:25 enantioselectivity (entry 5). Further introduction of substituents (e.g. Br and NO₂) on the indanol aromatic ring significantly improved the planar enantioselectivity, in which catalyst **G** furnished the product in 91:9 selectivity, albeit in a modest yield resulting from the competing intermolecular lactonization process to form the dimerization side product (see Supplementary Information for details) (entry 7). By evaluation of other catalysts, bases (entries 8–11) as well as various solvents and temperatures (see Supplementary Table 1, Supplementary Information for details), we were delighted to achieve the stereoselective synthesis of planar chiral product **2a** with 64% yield and 93:7 er by employment of

catalyst **G** with DABCO as the base (entry 11). Encouraged by the successful examples on cooperative NHC and co-catalysts[50–55], we then turned to examine various hydrogen bond donors (HBD), as well as Lewis acids/Brønsted acids to further enhance the catalytic performance (Supplementary Table 1, Supplementary Information for details). Satisfyingly, an optimal condition was obtained utilizing a cinchona-derived HBD-**3** as a cocatalyst in the presence of a mixed solvent (toluene:*n*-heptane), furnishing the product **2a** in 80% isolated yield and 96:4 er (entry 15). The improved results might arise from the additional hydrogen-bonding interaction provided by the thiourea moiety of HBD-**3** with the hydroxyl group and NHC-bound acyl azolium intermediate[31–36]. Furthermore, the use of enantiomeric NHC **G** (*ent*-**G**) as the carbene catalyst afforded the product **2a** in 70% yield and a reversed 6:94 er (entry 16), indicating a slight match/mismatch relationship between NHC **G** and co-catalyst HBD-**3**. It is

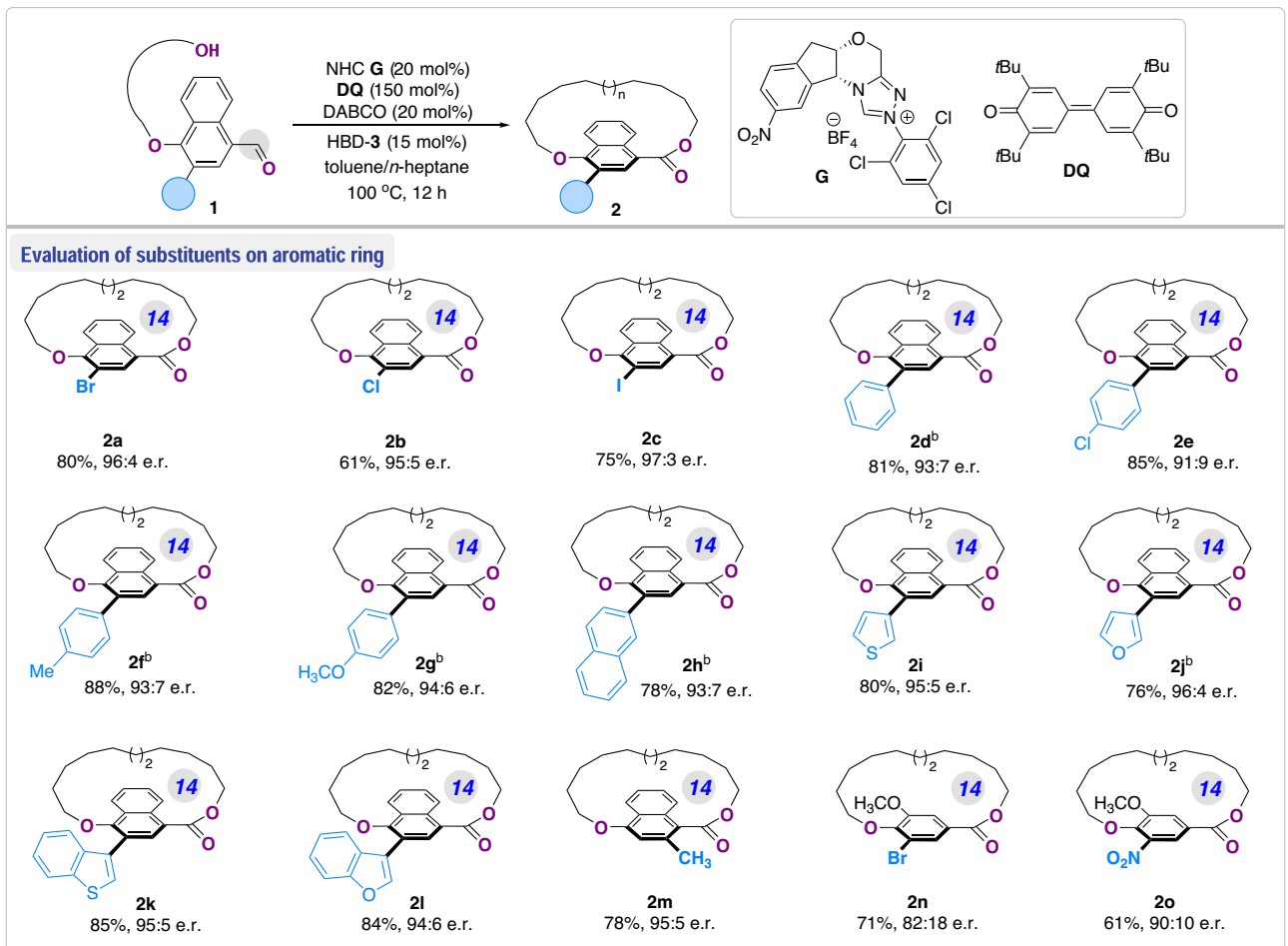

**Fig. 4 | Substrate scope of the enantioselective macrolactonization.** [a] The reactions were conducted with **1** (0.05 mmol, 1.0 equiv.), NHC **G** (5.2 mg, 20 mol%), DQ (30.6 mg, 150 mol%), HBD-**3** (5.2 mg, 15 mol%), and DABCO (1.12 mg, 20 mol%) in toluene/*n*-heptane (11:9 v/v, 1 mM) under N₂ atmosphere at 100 °C for 12 h; Isolated yields; [b] Reactions at 70 °C for 12 h.

also worth noting that the cyclophane product **2a** showed a remarkable configurational stability upon thermal racemization experiment of macrolactone **2a** in mesitylene, in which erosion of enantioselectivity was not observed even at 150 °C when the title compound **2a** started to decompose.

With the optimal reaction conditions established, we set out to study the generality of the carbene-catalyzed synthesis of planar chiral macrolactones (Fig. 4). An array of substituents on the aromatic moiety of aldehyde substrate **1** were initially explored in the catalytic stereo-selective macrocyclization. In addition to bromo group, substrates with chloro- and iodo-functional units were readily converted smoothly under the optimal conditions, delivering to the corresponding products **2b** and **2c** in 61–75% yields with high enantios-electivities (95:5 and 97:3 er, respectively). Installment of a diverse set of aromatic substitutions on the naphthalene moiety of **1** was subsequently studied. To our delight, the 3-phenyl substituted substrate afforded the macrolactone **2d** with 81% yield and 93:7 er. Various substituents such as Cl, CH₃, OCH₃ at the *para*-position of 3-phenyl group showed excellent compatibility under this condition, providing products **2e**–**g** with even higher yields (82–88%) and good stereo-selectivities. Furthermore, the 3-aryl unit on the naphthalene core of **1** could be replaced with 2-naphthyl (**2h**) and various heteroaromatic units such as thienyl (**2i**), furyl (**2j**), benzothienyl (**2k**) and benzofuranyl (**2l**) substituents, which significantly expanded the scope of planar chiral macrolactone derivatives. Further modification of the 3-substituent to 2-position of the naphthalene scaffold with a simple

methyl group furnished the product **2m** in 78% yield and 95:5 er. Noteworthy is that [18]-paracyclophanes **2n** and **2o** with *ortho*-disubstituted phenyl ring could also be prepared with our method, wherein the NO₂ group was compatible under the catalytic conditions albeit in a slightly dropped enantioselectivity (**2o**, 90:10 er). Notably, these planar chiral products exhibit substantial configurational stability as well, as demonstrated by product **2o**, which did not show decreased er value upon heating in mesitylene at 150 °C, while the title compound **2o** gradually decomposed.

Next, we turned to examine the length of the *ansa* chain to prepare paracyclophanes with various ring sizes (Fig. 5). Substrates with a 13-, or 12-membered *ansa* chain were readily converted to the corresponding planar chiral macrolactones **2p** (with a 17-membered macrocycle) and **2q** (with a 16-membered macrocycle) in excellent enantioselectivity. Notably, the absolute configuration of the planar chiral macrolactone products was established as (*R*ₚ)[56] by analogy to product **2p** via X-ray crystallographic analysis. However, further reducing the *ansa* chain to 11 members failed to give the corresponding product **2r** under the optimal conditions, most likely due to the unfavorable formation of a rigidified macrocycle. On the other hand, the side chain could be readily extended to 15 and 16 members, which produced the products **2s** and **2t** in 93:7 and 86:14 er, respectively. Furthermore, various functionalized linkers in the *ansa* chain, such as thioether (**2u**), 1,3-diyne (**2v**) and ether (**2w**), were compatible to deliver the corresponding products **2u**–**w** in modest yields and high enantioselectivities.

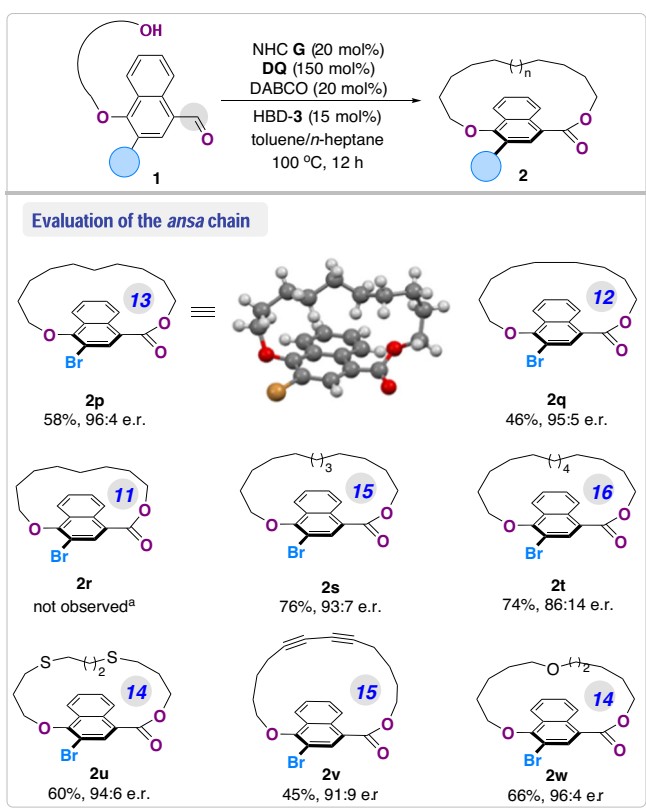

**Fig. 5 | Exploration of the *ansa* chains.** [a] The reactions were conducted with **1** (0.05 mmol, 1.0 equiv.), NHC **G** (5.2 mg, 20 mol%), DQ (30.6 mg, 150 mol%), HBD-**3** (5.2 mg, 15 mol%), and DABCO (1.12 mg, 20 mol%) in toluene/*n*-heptane (11:9 v/v, 1 mM) under N₂ atmosphere at 100 °C for 12 h; Isolated yields.

The optically enriched planar chiral macrolactones prepared in our approach could readily undergo further synthetic transformations (Fig. 6). A palladium-catalyzed Suzuki cross coupling reaction between **2a** and **6** afforded chiral cyclophane **7** with 80% yield and 94:6 er. Other transition-metal-catalyzed couplings were also viable to diversify the catalytically obtained macrolactone products. For instance, a Heck reaction with styrene **8** enabled by Pd(OAc)₂/PPh₃ afforded the alkene-tethered planar chiral macrocycle **9** in 42% yield and 95:5 er. Additionally, Sonogashira coupling of **2c** with terminal alkyne **10** led to product **11** in 92% yield and without erosion of er value.

In summary, we have developed a carbene organocatalytic approach for planar enantioselective macrolactonization. A wide range of cyclophanes, featuring intriguing configurationally stable planar stereogenicity owing to the restricted ring flip of the macrocycles, were obtained efficiently in high yields and excellent stereoselectivities under oxidative NHC conditions. Diversification of the chiral macrocycles were readily achieved through a series of coupling reactions to significantly expand the scope of this method. Furthermore, our approach provides a (non-enzymatic) organocatalytic approach to address the long-standing challenge in stereoselective preparation of planar chiral macrolactones. New avenues by carbene organocatalytic approach for synthetic implementation to access optically enriched planar chiral frameworks and biologically intriguing macrocyclic scaffolds could be anticipated. Ongoing studies in our laboratory include development of the prepared planar chiral macrolactones for chiral catalyst design, and biological activity evaluation for novel agrochemical discovery.

## Methods

To a 100.0 mL Schlenk flask equipped with a magnetic stir bar was added chiral NHC pre-catalyst **G** (5.2 mg, 20 mol%), DQ (30.6 mg, 150 mol%,), DABCO (1.12 mg, 20 mol%), HBD-**3** (5.17 mg, 15 mol%) and

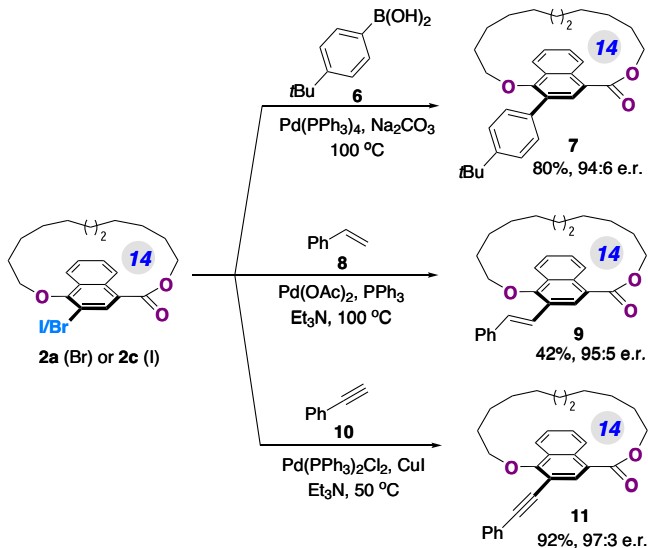

**Fig. 6 | Synthetic transformations of the obtained products 2a and 2c.** Detailed reaction conditions for the transformations are presented in the Supplementary Information.

aldehyde substrate **1** (0.05 mmol, 1.0 equiv.). After that, a mixed solvent of toluene / *n*-heptane (11:9 *v/v*, 1 mM) was added and the reaction mixture was allowed to stir for 12 h at 100 °C. Then the mixture was concentrated under reduced pressure. The resulting crude residue was purified by column chromatography on silica gel to afford the desired planar chiral product **2**.

## Data availability

The X-ray crystallographic coordinates for structures of the compounds ($R_p$)-**2p** reported in this study have been deposited at the Cambridge Crystallographic Data Centre (CCDC), under deposition numbers **CCDC 1849177**. These data can be obtained free of charge from The Cambridge Crystallographic Data Centre via www.ccdc. cam.ac.uk/data_request/cif. The full experimental details for the preparation of all new compounds, and their spectroscopic and chromatographic data generated in this study are provided in the Supplementary Information/Source Data file. All data are available from the authors upon request.

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

## Acknowledgements

We acknowledge funding supports from the National Natural Science Foundation of China (21732002, 22061007, 22071036, and 22207022, Y.R.C.); Frontiers Science Center for Asymmetric Synthesis and Medicinal Molecules, National Natural Science Fund for Excellent Young Scientists Fund Program (Overseas)-YQHW (Z2023776, X.W.), the starting grant of Guizhou University [(2022)47], X.W.], Department of Education, Guizhou Province [Qianjiaohe KY number (2020)004, Y.R.C.]; The 10 Talent Plan (Shicengci) of Guizhou Province ([2016] 5649, Y.R.C.); Science and Technology Department of Guizhou Province [Qiankehe-jichu-ZK[2022]zhongdian024, Y.R.C.], ([2018]2802, [2019]1020, Y.R.C.), QKHJC-ZK[2022]-455, Y.R.C.; Department of Education of Guizhou Province (QJJ(2022)205, Y.R.C.); Program of Introducing Talents of Discipline to Universities of China (111 Program, D20023, X.W., Y.R.C.) at Guizhou University; Singapore National Research Foundation under its NRF Investigatorship (NRF-NRFI2016-06, Y.R.C.) and Competitive Research Program (NRF-CRP22-2019-0002, Y.R.C.); Ministry of Education, Singapore, under its MOE AcRF Tier 1 Award (RG7/20, RG70/21, Y.R.C.), MOE AcRF Tier 2 (MOE2019-T2-2-117, Y.R.C.), and MOE AcRF Tier 3 Award (MOE2018-T3-1-003, Y.R.C.); a Chair Professorship Grant, and Nanyang Technological University.

## Author contributions

X.L. performed main methodology development, scop evaluation and synthetic application; F.S., H.L., F.L. contributed to earlier studies; Y.Z., M.L. and F.C. contributed to scope evaluation and synthetic application; X.W. and Y.R.C. conceptualized and directed the project and drafted the manuscript with assistance from all co-authors. All authors contributed to part of the experiments and/or discussions.

## Competing interests

The authors declare no competing interests.
