## [Peer Review File · Nature Communications]

Carbene Organic Catalytic Planar Enantioselective MacrolactonizationReviewers' Comments:

Reviewer #1:

Remarks to the Author:

Macrolactones are frequently found not only in nature but also in medicines. Despite the great efforts made for their synthesis, challenges still exist in the exploration of efficient methods to govern both lactone formation and stereochemical control. Chi, Wu, and co-workers reported an NHC-catalyzed intramolecular macrocyclization of aldehydes for the planar enantioselective construction of macrolactones with 16-20 ring size. This work opens a new window for accessing macrolactones with planar chirality by organocatalysis. The substrate scope was carefully explored and manuscript was well organized. Basically, the publication of this work in Nature Communication is recommend, and some issues need to be addressed before publication.

1. NHC-catalyzed reactions rarely need such a high temperature (100oC for this reaction) which might lead to complexity of the reaction and difficulty in the control of the stereochemistry. Can the authors give some comments on this result?
2. Are there any intermolecular lactone products observed in this reaction system?
3. Is it possible to replace the oxygen with other linkers such as sulfur or nitrogen?
4. In SI, there are many impurities in the spectra of substrate 1. The spectra of 2k and 11 are not clean.

Reviewer #2:

Remarks to the Author:

In the manuscript, Wu and Chi et al. report the enantioselective synthesis of planar chiral macrolactones via NHC-catalyzed intramolecular lactonization of alcohol-tethered naphthol aldehydes. Macrolactones with rings from sixteen to twenty members are obtained in good yields with good to high enantioselectivities. There are very few methodology reported for the construction of planar chiral macrolactones with high enantioselectivity. The work is remarkable and high potentially useful. The publication in Nat. Comm. is recommended after revisions.

- (1) Both the NHC and HBD additive are chiral. The match or mismatch issue of the chiral NHC G and HBD-3 should be clarified?
- (2) If possible, provide the stereocontrol mode(s).
- (3) How is the enantioselectivity, if the 2-substituent is re-located at 3-position. What about substrates with 2,3-disubstutents.
- (4) What about this NHC catalysis employed for the reaction of dicarboxylic acid with diols, such as the reaction in Figure 1c?

Response to referee comments:

Referee #1 Macrolactones are frequently found not only in nature but also in medicines. Despite the great efforts made for their synthesis, challenges still exist in the exploration of efficient methods to govern both lactone formation and stereochemical control. Chi, Wu, and co-workers reported an NHC-catalyzed intramolecular macrocyclization of aldehydes for the planar enantioselective construction of macrolactones with 16-20 ring size. This work opens a new window for accessing macrolactones with planar chirality by organocatalysis. The substrate scope was carefully explored and manuscript was well organized. Basically, the publication of this work in Nature Communication is recommend, and some issues need to be addressed before publication.

Our response: We would first like to thank referee 1 for the very constructive comments, the recommendation for publication and the insightful suggestions for improving our manuscript.

1) Referee's comment: NHC-catalyzed reactions rarely need such a high temperature (100 °C for this reaction) which might lead to complexity of the reaction and difficulty in the control of the stereochemistry. Can the authors give some comments on this result?

Our response: During the optimization of the reaction conditions, we carefully studied the influence of the temperature on the reaction outcome (additional conditions screening on the temperature were added to Table S2 of the revised Supporting Information).

We found that the temperature showed a significant influence on the product yields, whereas no apparent effect on the stereo-control. Low temperature (<80 °C) would lead to a complex mixture, such as dimer, trimer or other side products. Considering the challenging ring closure owing to the rigidified macrocycle, elevated temperatures (e.g. 100 °C) were used to promote the intramolecular macrolactonization, giving the desired product in acceptable yields. In addition, our intramolecular reaction employed a relative dilute concentration (10^{-3} M) and moisture- and oxygen-free reaction conditions, which might also reduce the potential unproductive side reaction process when at high temperature, such as carbene deactivation, hydrolyzation, oxidation to acid by O₂ etc.

Table S2. Examination of the effects of reaction temperatures^a

Entry	Temp.	Yield (%) ^b	E.r. (%)
1	25	37	95:5
2	50	33	95:5
3	70	42	96:4
4	100	82	96:4
5	110	76	96:4

^a General conditions (unless otherwise specified): **1a** (6.3 mg, 0.015 mmol, 1.0 equiv.), NHC **G** (20 mol%), **DQ** (9.2 mg, 150 mol%), and DABCO (20 mol%) in toluene/*n*-heptane (11:9 v/v, 1 mM) under N₂ atmosphere at indicated temperature for 12-24 h (entries 1-3, 24 h; entries 4-5, 12 h). ^b Yields of **2a** were determined via ¹H NMR analysis with 1,3,5-trimethoxybenzene as an internal standard.

2) Referee's comment: Are there any intermolecular lactone products observed in this reaction system?

Our response: Yes, during the optimization of the reaction conditions, the intermolecular lactonization was the major unproductive reaction pathway. We engaged extensive efforts, by screening the mixed solvents, concentration of the starting material and the use of additives, to suppress this side reaction process. To our delight, with the developed optimal conditions, we finally could obtain the desired products in acceptable yields and enantioselectivities. We have added a text "albeit in a modest yield resulting from the competing intermolecular lactonization process to form the dimerization side product (see Supporting Information for details)" directly to the optimization section of the revised manuscript (Page 3 of the revised manuscript, highlighted in yellow). Accordingly, the Supporting Information has also been revised accordingly (Page S13, of the revised SI)

For characterization, see: Page S13, of the revised SI.

3) Referee's comment: Is it possible to replace the oxygen with other linkers such as sulfur or nitrogen?

Our response: We are grateful for this suggestion and have prepared the corresponding substrates **S4a** and **S5a** (see Page S14-S19 of the revised Supporting Information). However, after extensive investigations, we found that these two types of substrates were not compatible in our developed macrocyclization. Furthermore, we have also explored other linkers in the ansa chain, instead of at the terminal position, such as thioether (**1u**), 1,3-diynes (**1v**) and ether (**1w**), which successfully afforded the corresponding products **2u-w** in high enantioselectivity. The details of these studies were given as below.

-Firstly, we attempted to obtain the macrocyclization product **S4b** with S-based precursor **S4a**. Initial conditions screening with the optimal conditions failed to give any desired product **S4b** (Table S3, entry 1). We then performed the reaction at elevated temperatures at 110-150 °C (Table S3, entries 2-4). However, we were still not able to observe any product **S4b**, with a large amount of starting material remained in the reaction. Furthermore, we tested additional reaction conditions with various NHC catalysts, solvents and temperatures (Table S4, on Page S16 of the revised Supporting Information). While the product **S4b** was still not obtained, a side product **S4c** was isolated in several entries.

Table S3. Macrolactonization of thiol-based substrate **S4a** with NHC **G**

Entry	Solvent	Temp. (°C)	Results
1	Toluene/ n -Heptane	100	A large amount of S4a remained; S4b not observed
2	Toluene/ n -Heptane	110	A large amount of S4a remained; S4b not observed
3	PhCl	130	A large amount of S4a remained; S4b not observed

^a General conditions (unless otherwise specified): **S4a** (6.5 mg, 0.015 mmol, 1.0 equiv.), NHC **G** (20 mol%), **DQ** (9.2 mg, 150 mol%), and DABCO (20 mol%) in solvents (1 mM; entries 1-2, toluene/*n*-heptane = 11:9 v/v) under N₂ atmosphere at indicated temperature for 12 h.

Table S4. Additional screening on the macrolactonization with substrate **S4a**

Entry	NHC	Base	Solvent	Temp.	Results
1	A, B, D, K, J	K ₂ CO ₃	THF	70	A large amount of S4a remained
2	L	K ₂ CO ₃	THF	70	S4c was obtained
3	B, D, J, K	K ₂ CO ₃	Toluene	100	A large amount of S4a remained
4	A, L	K ₂ CO ₃	Toluene	100	S4c was obtained
5	A, B, D, K	Cs ₂ CO ₃	THF	70	A large amount of S4a remained
6	J, L	Cs ₂ CO ₃	THF	70	S4c was obtained
7	A, B, D, J, L	Cs ₂ CO ₃	Toluene	100	A large amount of S4a remained
8	K	Cs ₂ CO ₃	Toluene	100	S4c was obtained

^a General conditions (unless otherwise specified): **S4a** (6.5 mg, 0.015 mmol, 1.0 equiv.), NHC Cat. (20 mol%), **DQ** (9.2 mg, 150 mol%), and base (20 mol%) in solvents (1 mM) under N₂ atmosphere at indicated temperature for 12 h.

-Secondly, following the suggestions, we studied the macrocyclization with N-based precursor **S5a**. The starting material **S5a** was successfully prepared. However, similar to the reaction with precursor **S4a**, we could not obtain the corresponding product **S5b** with our catalytic conditions or at elevated temperatures (Table S5, entries 1-4, on Page S18 of the revised Supporting Information). Additional screenings with NHCs, solvents and temperatures also failed to give the desired macrocyclization product **S5b** (Table S6). Coupling product **S5c** with the reduced hydroquinone was detected in several reactions.

Table S5. Macrolactonization of thiol-based substrate **S4a** with NHC **G**
Entry	Solvent	Temp. (°C)	Results
1	Toluene/ n -Heptane	100	no S5b was observed
2	Toluene/ n -Heptane	110	no S5b was observed
3	PhCl	130	no S5b was observed
4	Mesitylene	150	no S5b was observed

^a General conditions (unless otherwise specified): **S5a** (8.6 mg, 0.015 mmol, 1.0 equiv.), NHC **G** (20 mol%), **DQ** (9.2 mg, 150 mol%), and DABCO (20-100 mol%) in solvents (1 mM; entries 1-2, toluene/*n*-heptane = 11:9 v/v) under N₂ atmosphere at indicated temperature for 12 h.

Table S6. Additional screening on the macrolactonization of substrate **S5a**
Entry	NHC	Base	Solvent	Temp.	Results
1	A, B, K	Cs ₂ CO ₃ or K ₂ CO ₃	THF	70	A large amount of S5a left
2	D, J, L	Cs ₂ CO ₃ or K ₂ CO ₃	THF	70	S5c was obtained
3	A, B, K	Cs ₂ CO ₃ or K ₂ CO ₃	Toluene	100	A large amount of S5a left
4	D, J, L	Cs ₂ CO ₃ or K ₂ CO ₃	Toluene	100	S5c was obtained

^a General conditions (unless otherwise specified): **S5a** (8.6 mg, 0.015 mmol, 1.0 equiv.), NHC (20 mol%), **DQ** (9.2 mg, 150 mol%), and base (20-100 mol%) in solvents (1 mM) under N₂ atmosphere at indicated temperature for 12 h.

- Lastly, we have also explored other linkers in the ansa chain, instead of at the terminal position, such as thioether (**1u**), 1,3-diynes (**1v**) and ether (**1w**). The reactions afforded the corresponding products **2u-w** in high enantioselectivity. A text was added to the revised manuscript, "Furthermore, various functionalized linkers in the ansa chain, such as thioether (**2u**), 1,3-diyne (**2v**) and ether (**2w**), were compatible to deliver

the corresponding products **2u-w** in modest yields and high enantioselectivities.” on Page 6 of the revised manuscript, highlighted in yellow. The characterization of these products were also added to the revised SI (See Page S59-S60, and Page S137-S142).

Fig. 5

4) Referee's comment: In SI, there are many impurities in the spectra of substrate 1. The spectra of **2k** and **11** are not clean.

Our response: Following the suggestions, we have repurified the compounds and revised the spectra of substrates **1h**, **1k**, **1l**, **1r**, **1t** and products **2o**, **2k**, **11** accordingly (see: substrates **1h** on Page S82, **1k** on Page S85, **1l** on Page S86, **1r** on Page S92, **1t** on Page S94, as well as products **2o** on Page S127, **2k** on Page S119 and **11** on Page S147.)

Referee #2 In the manuscript, Wu and Chi et al. report the enantioselective synthesis of planar chiral macrolactones via NHC-catalyzed intramolecular lactonization of alcohol-tethered naphthol aldehydes. Macrolactones with rings from sixteen to twenty members are obtained in good yields with good to high enantioselectivities. There are very few methodology reported for the construction of planar chiral macrolactones with high enantioselectivity. The work is remarkable and high potentially useful. The publication in Nat. Comm. is recommended after revisions.

Our response: We are truly grateful to referee 2 for the enthusiasm, the very helpful comments and the thoughtful assessment of our manuscript.

1) Both the NHC and HBD additive are chiral. The match or mismatch issue of the chiral NHC **G** and HBD-3 should be clarified?

Our response: We are grateful for the referee's thoughtful suggestions. We have performed the control experiments with the *ent*-NHC **G** and thiourea HBD-3 as the catalyst combinations. The reaction gave the product **2a** in 70% yield and a reversed 6:94 er (results were added to Fig. 3, entry 16). In comparison to the results of **2a** (80% yield and 96:4 er) under the optimal conditions (as in entry 15, Fig. 3), the results implicate a slight match/mismatch relationship between NHC **G** and HBD 3 co-catalyst. (a text was added to Page 4 of the revised manuscript, highlighted in yellow: "Furthermore, the use of enantiomeric NHC **G** (*ent*-**G**) as the carbene catalyst afforded the product **2a** in 70% yield and a reversed 6:94 er (entry 6), indicating a slight match/mismatch relationship between NHC **G** and HBD 3 co-catalyst.")

Entry	Cat.	Base	Additive	Yield, 2a (%)	E.r.
15	G	DABCO	HBD 3	82 (80)	96:4
16	ent - G	DABCO	HBD 3	70	6:94

Fig. 3 (Page 4 of the revised manuscript)

2) If possible, provide the stereocontrol mode(s).

Our response: We agree and have included the plausible mechanism with the stereocontrol modes to account for the obtained planar chiral products in the developed macrocyclizations. (see Scheme S4, Page S12 of the revised Supporting Information)

Scheme S4, Page S12 of the revised SI

3) How is the enantioselectivity, if the 2-substituent is re-located at 3-position. What about substrates with 2,3-disubstutents.

Our response: Following this valuable recommendation, we have synthesized the corresponding substrates **S6a** and **S7a** for further studies, with results given as below.

- In our scope, we have tested a 3-methyl substituted substrate **1m** that could afford the corresponding product **2m** in 78% yield and 95:5 er. However, when 3-Cl substituted precursor **S6a** was subjected to the optimal conditions, no any corresponding product **S6b** was observed with our catalytic conditions or at elevated temperatures (Table S7, entries 1-4, on Page S21 of the revised Supporting Information). Additional screenings with NHCs, solvents and temperatures also failed to give the desired macrocyclization product **S6b** (Table S8), wherein particular conditions would give the dimer/trimer side products **S6c** and **S6d**, probably owing to the notable enhanced rigidified macrocyclic structure of **S6b** that poses difficulty on the intramolecular ring closure.

Table S7. Macrolactonization of substrate S6a with NHC G

Entry	Solvent	Temp. (°C)	Results
1	Toluene/ n -Heptane	100	S6a remained unreacted
2	Toluene/ n -Heptane	110	S6a remained unreacted
3	PhCl	130	S6a remained unreacted
4	Mesitylene	150	S6a remained unreacted

^a General conditions (unless otherwise specified): **S6a** (5.6 mg, 0.015 mmol, 1.0 equiv.), NHC **G** (20 mol%), **DQ** (9.2 mg, 150 mol%), and DABCO (20 mol%) in solvents (1 mM; entries 1-2, toluene/*n*-heptane = 11:9 v/v) under N₂ atmosphere at indicated temperature for 12 h.

Table S8. Additional screening on the macrolactonization with substrate S6a

Entry	NHC	Base	Solvent	Temp.	Result
1	A, B, D, K, L	K ₂ CO ₃	THF	70	A large amount of S6a remained
2	J	K ₂ CO ₃	THF	70	S6c and S6d was isolated; product S6b was not observed
3	B, D, J-L	K ₂ CO ₃	Toluene	100	A large amount of S6a remained
4	A	K ₂ CO ₃	Toluene	100	S6c and S6d was isolated; product S6b was not observed
5	A, B, D, J	Cs ₂ CO ₃	THF	70	A large amount of S6a remained
6	K, L	Cs ₂ CO ₃	THF	70	S6c and S6d was isolated; product S6b was not observed
7	A, B, D, J	Cs ₂ CO ₃	Toluene	100	A large amount of S6a remained
8	K, L	Cs ₂ CO ₃	Toluene	100	S6c and S6d was isolated; product S6b was not observed

^a General conditions (unless otherwise specified): **S6a** (5.6 mg, 0.015 mmol, 1.0 equiv.), NHC (20 mol%), **DQ** (9.2 mg, 150 mol%), and base (20 mol%) in solvents (1 mM; entries 1-2, toluene/*n*-heptane = 11:9 v/v) under N₂ atmosphere at indicated temperature for 12 h.

- Secondly, a 2,3-disubstituted precursor **S7a** was studied for the macrolactonization. Despite extensive conditions have been screened, such as with various NHCs, solvents and temperatures, we could not obtain the corresponding product **S7b** (Table S9 and S10, Page S25-S27 of the revised Supporting Information). In these cases, the starting material **S7a** were all remained unreacted, no any dimer/trimer product could be observed, which indicates that the carbene addition to the aldehyde most likely not occurred smoothly to give the key acyl azolium species due to the steric hinderance of the multisubstituted arylcarboxyaldehyde **S7a**.

Table S9. Macrolactonization of substrate **S7a** with NHC **G**

Entry	Solvent	Temp. (°C)	Results
1	Toluene/ n -Heptane	100	S7a remained unreacted
2	Toluene/ n -Heptane	110	S7a remained unreacted
3	PhCl	130	S7a remained unreacted
4	Mesitylene	150	S7a remained unreacted

^a General conditions (unless otherwise specified): **S7a** (6.5 mg, 0.015 mmol, 1.0 equiv.), NHC **G** (20 mol%), **DQ** (9.2 mg, 150 mol%), and DABCO (20 mol%) in solvents (1 mM; entries 1-2, toluene/*n*-heptane = 11:9 v/v) under N₂ atmosphere at indicated temperature for 12 h.

Table S10. Additional screening on the macrocyclization with substrate **S7a**

Entry	NHC	Base	Solvent	Temp.	Result
1	A, B, D, J, M, N	K ₂ CO ₃	THF	70	S7a remained unreacted
2	A, B, D, J, M, N	Cs ₂ CO ₃	THF	70	S7a remained unreacted
3	A, B, D, J, M, N	K ₂ CO ₃	Toluene	100	S7a remained unreacted
4	A, B, D, J, M, N	Cs ₂ CO ₃	Toluene	100	S7a remained unreacted

^a General conditions (unless otherwise specified): **S7a** (6.5 mg, 0.015 mmol, 1.0 equiv.), NHC (20 mol%), **DQ** (9.2 mg, 150 mol%), and base (20 mol%) in solvents (1 mM) under N₂ atmosphere at indicated temperature for 12 h.

4) What about this NHC catalysis employed for the reaction of dicarboxylic acid with diols, such as the reaction in Figure 1c?

Our response: We are grateful for the referee's insightful suggestions. The previous elegant report by Collins et al. achieved the enzyme-catalyzed direct intermolecular macrocyclization with dicarboxylic acid and diols as substrates. We applied our carbene catalytic conditions to Collin's reaction system, with acid **S8** and diol **S9** as the two model substrates. In contrast to the DQ oxidative conditions, carboxylic acid in-situ

activation with acyl chloride was used to give the corresponding key acyl azolium intermediate. From our investigations as illustrated in Table S11, we were delighted to find that the reaction with NHC **G** in toluene readily afforded the chiral product **S10** in 31% yield and 59:41 er, as a proof-of-concept for the carbene-catalyzed intermolecular macrolactonization strategy for further exploration.

Table S11. Investigation on the intermolecular macrolactonization between **S8** and **S9**^a

Entry	NHC	Solvent	Temp. (°C)	Yield (%) ^b	E.r. (%)
1	A	CH ₂ Cl ₂	40	28	50:50
2	J	CH ₂ Cl ₂	40	33	50:50
3	B	CH ₂ Cl ₂	40	30	50:50
4	C	CH ₂ Cl ₂	40	0	n.d.
5	D	CH ₂ Cl ₂	40	32	50:50
6	A-D, J	THF	70	0	n.d.
7	A	Toluene	50	0	n.d.
8	A	Toluene	100	29	50:50
9	B	Toluene	100	20	56:44
10	C	Toluene	100	25	52:48
11	D	Toluene	100	30	50:50
12	G	Toluene	100	31	59:41
13	O	Toluene	100	25	55:45
14	P	Toluene	100	21	48:52

^a General conditions (unless otherwise specified): **S8** (4.2 mg, 0.01 mmol, 1.0 equiv.), **S9** (1.7 mg, 0.01 mmol, 1.0 equiv.), NHC (20 mol%), PivCl (2.9 mg, 240 mol%), and DIPEA (6.2 mg, 480 mol%) in solvent (0.1 M) at indicated temperature for 12 h. ^b Yields of **2a** were determined via ¹H NMR analysis with 1,3,5-trimethoxybenzene as an internal standard.

Editorial office

1) Please complete or update the following checklist(s) to verify compliance with our research ethics and data reporting standards. Address all points on the checklist, revising your manuscript in response to the points if needed.

The form(s) must be downloaded and completed in Adobe Reader rather than opened in a web browser. Each form must be uploaded as a Related Manuscript file at the time of resubmission.

Our response: We have filled the form and uploaded it in the submission system.

2) Your work characterises chemical or biomolecular materials. Please see the link below for reporting requirements. There is no form to upload but you may need to revise your manuscript to comply with this policy.

Our response: Not relevant to our research.

3) DATA AND CODE AVAILABILITY* All Nature Communications manuscripts must include a “Data Availability” section after the Methods section but before the References. If any of the data can only be shared on request or are subject to restrictions, please specify the reasons and explain how, when, and by whom the data can be accessed.

Our response: We have included the Data Availability” section before the References. (“All the data supporting the findings of this study are available within the article and its Supplementary Information file. All other data are available from the corresponding author Xingxing Wu or Yonggui Robin Chi.” on Page S7 of the revised manuscript, highlighted in yellow.)

Reviewers' Comments:

Reviewer #1:

Remarks to the Author:

The authors have well responded to the issues, and the current form could be accepted for publication.

Reviewer #2:

Remarks to the Author:

The concerns from the reviewers are fully addressed. The publication in Nat. Commun. is recommended.